# In Situ Experimental Study of Cloud-Precipitation Interference by Low-Frequency Acoustic Waves

**Yang Shi** [1,2,†], **Zhen Qiao** [1,†], **Guangqian Wang** [1,2] **and Jiahua Wei** [1,2,*]

1   State Key Laboratory of Hydroscience and Engineering, Tsinghua University, Beijing 100084, China
2   State Key Laboratory of Plateau Ecology and Agriculture, Laboratory of Ecological Protection and High-Quality Development in the Upper Yellow River, School of Water Resources and Electric Power, Qinghai University, Xining 810016, China
*   Correspondence: weijiahua@tsinghua.edu.cn; Tel.: +86-10-6279-6325
†   These authors contributed equally to this work.

**Abstract:** Since acoustic agglomeration is an effective pre-treatment technique for removing fine particles, it can be considered as a potential technology for applications in aerosol pollution control, industrial dust and mist removal, and cloud and precipitation interference. In this study, the cloud-precipitation interference effect was evaluated in situ based on a multi-dimensional multi-scale monitoring system. The variations in the spatial and temporal distribution of rainfall near the surface and the characteristics of precipitation droplets in the air were investigated. The results indicate that strong low-frequency acoustic waves had a significant impact on the macro-characteristics of rainfall clouds, the microphysical structure of rain droplets and near-surface precipitation, and various microwave parameters. In terms of physical structure, the precipitation cloud's base height decreased significantly upon opening the acoustic device, while agglomeration and de-agglomeration of raindrops were in a dynamic equilibrium. When the sound generator was on, the particle concentration at a sampling attitude of $500-1700$ m and the proportion of particles with diameters of 1–1.5 mm decreased significantly (by 1–5 ln [$1/m^3 \cdot mm$]). In contrast, the particle concentration increased by 1–3 ln [$1/m^3 \cdot mm$] at a sampling attitude below 400 m. Moreover, during acoustic interference, the reflectivity factor surged by 2.71 dBZ within 1200 m of the operation centre. Overall, the spatial and temporal distributions of rainfall rates and cumulative precipitation within 5 km of acoustic operation were uneven and influenced by local terrain and background winds.

**Keywords:** meteorology; acoustic interference of atmosphere; physical examination; atmospheric water exploitation; cloud microphysical characteristics; source region of the Yellow River (SRYR)

## 1. Introduction

China has abundant air water resources, with an annual water vapour input of approximately 22.6 trillion cubic meters [1]. However, there is a significant imbalance in the spatial and temporal distribution of precipitation, and the conversion rate varies widely depending on the region [2,3]. Many areas face problems related to perennial or seasonal water scarcity [3,4]. A growing understanding of the physical process of cloud precipitation enables higher cloud-precipitation conversion rates and the development of air water resources [5–9].

Cloud and precipitation interference, which is the main form of atmospheric water exploitation, has been an attractive research area [10–12]. It involves stimulating or accelerating nucleation and precipitation processes by changing the colloidal steady state in clouds [13–15]. The most commonly used method is to alter the phase state and spectral distribution of cloud droplets by seed catalysts [16], such as dry ice or silver iodide, in an appropriate weather and geographical environment [7,17], using vehicles such as aircraft [18], unmanned aerial vehicles (UAVs) [7,17,19–21], and artillery [22–24]. This promotes the microphysical and macroscopic dynamic processes of cloud formation and

precipitation [25]. However, the long-term use of chemicals may introduce ecological and environmental pollution risks. $Ag^+$ concentrations in lakes and rivers [26,27], soils [28,29], and the atmosphere [30,31] after catalytic precipitation have been continuously studied since the 1960s [32], but their toxic influences are still controversial [33,34]. Meanwhile, the use of ground artillery and aircraft to disperse catalysts requires airspace permission, and the cost of carriers is expensive. Therefore, it is crucial to develop eco-friendly and efficient technologies for cloud and precipitation interference.

Recently, numerous approaches such as acoustic agglomeration [6,35], laser-induced coagulation [36,37], and atmospheric ionization [38,39], have emerged and attracted more attention. Among these, acoustic agglomeration is already widely used in industrial dust and mist removal [40,41] and is considered to have potential applications in precipitation enhancement due to its catalyst-free, simple, and reliable characteristics as well as low cost [42–44]. The comparison between acoustic intervention and the traditional methods can be found in the Supplementary Materials. The principle of acoustic precipitation is to cause air oscillations by emitting directional low-frequency and high-intensity sound waves toward the cloud, thereby disturbing the air in the acoustic field [35]. This increases the probability of collisions between droplets and ice crystals, which leads to the growth of condensation nuclei and ultimately precipitation [44–46]. Wei et al. [6] proposed that acoustic waves have a trigger and periodic effect on clouds in terms of radar echo intensity. Qiu et al. [47] demonstrated that acoustic waves intervene significantly for heavy precipitation systems with large rain rate ($R$), especially for $R > 10.42$. Wang et al. [48] proposed that clouds with larger precipitation potential are significantly beneficial to the acoustic coagulation process. The average rain rate can be increased by 72% subject to low-frequency acoustic waves during a two-hour rainfall event.

The assessment of acoustic interference has been demonstrated by a series of indoor-scale experiments and numerical simulations [14,15,49,50]. The principal mechanisms of acoustic agglomeration are orthokinetic and hydrodynamic interactions between droplet aerosols. Hoffmann and Koopmann [50] experimentally investigated orthokinetic interactions between polydisperse droplets under the actions of a strong acoustic field using a high-resolution CCD camera system. The hydrodynamic interaction mechanisms acoustically induced between monodisperse aerosols have been proven by González et al. [51,52]. Microscopic visualizations are direct proof of acoustic agglomeration, but most of the tests are kinematic responses of single particles and particle pairs, which are not consistent with polydisperse particle systems [51,53]. Additionally, there are some statistical studies on droplet agglomeration at the laboratory scale, while pressure and temperature are hardly consistent with those in the precipitous. The environmental parameters of operating pressure and ambient temperature may affect the transient flow process and the interaction between air flow and discrete droplets and thereby indirectly affect acoustic agglomeration [15]. For the numerical model, the microphysical and dynamic processes of droplet aerosols involved in artificial interferences can be described under a wide range of dynamic conditions and particle spectra. The reliability of this approach depends on the theory and observation of acoustic agglomeration [54]. However, existing studies on the migration and agglomeration processes of droplet aerosols have approximated them as sparse two-phase flows [55–57] and ignored the turbulence structure and interphase coupling effects [56,58,59]. This is undoubtedly inconsistent with the mechanism of particle multidynamic action in realistic environments, and it is unknown whether the obtained results are of engineering guidance. Therefore, it is necessary to further explore the macro- and micro-processes of cloud-precipitation under acoustic interference based on fine meteorological observations [60].

Field-scale experiments may be based on various detection techniques (e.g., direct or remote sensing and tracing) to confirm variations in the microphysical and dynamic responses of cloud-precipitation systems. To date, numerous field tests have been carried out to investigate the effects of artificial interference on clouds and precipitation using catalysts, acoustic waves and charged aerosols. Examples include randomized topographic cumulus

cloud seeding experiments in the Arizona Mountains, USA (1955–1964) [61]; nontopographic cumulus cloud seeding experiments in southern Missouri, USA (White Top Project, 1960–1964) [62,63]; random seed catalyst testing in Israel (between 1961 and 1991) [64]; winter storm intervention program in the Cascade Mountains, Washington, DC, USA (1969–1974) [65]; catalytic experiments in the Camagüey region of Cuba (1986–1990) [66]; artificial interference on cumulus clouds over southern Israeli cities (1976–1994) [67]; Colorado orographic cloud seeding experiments (COSE; 1979–1985) [68]; randomized seeding experiments at the Gutian Reservoir, Fujian, China (1975–1986) [69]; Australian Winter Storm Experiment (AWSE–I; 1988–1993) [70]; cloud and precipitation catalysis experiments using charged particles, Ningxia, China (2016–2021); and randomized acoustic interference on clouds and precipitation at Darlag and Delingha, Qinghai, China (2017–2021) [6,47]. The above tests provide physical evidence that can be used to assess the effects of artificial interference [2,11,12,71]. The near-ground rainfall distributions [5] and vertical microphysical characteristics of precipitation [72] under the action of low-frequency acoustic waves were monitored by rain gauges, micro rain radars, and microwave radiometers. However, research regarding the physical response of rain droplets and precipitation systems following artificial interferences often uses single or dual sources of data for analysis because of certain limitations of advanced cloud precipitation detection devices [18,73,74]; analysis using three or more observations is rare, especially for acoustic interference. To the best of our knowledge, there have not been any systematic experimental investigations of acoustic interference effects under natural weather conditions based on fine-scale detection and dynamic monitoring using multiple spatially and temporally matched monitoring data [75,76]. Qiu et al. [47] and Shi et al. [5] focused on surface rainfall distribution with acoustic intervention through rain gauges. Wei et al. [5] and Shi et al. [72] highlighted the variations in radar reflectivity in the zenith direction of the field site. The perspectives of the above studies are relatively simple, which somewhat limits the exploration of intervention patterns and performance optimization.

Rainfall is a complex weather process that involves the macro-movement of water vapour and strong air flow, as well as micro-dynamic behaviours, such as the collision, growth, and condensation of clouds and rain droplets. Since the spatiotemporal evolution of precipitating clouds has a large degree of natural variation and uncertainty [5,8,75], multidimensional meteorological exploration is required to understand the formation, development and disappearance processes of precipitation systems and to enhance interference efforts [71]. To date, there are few studies that systematically describe the evolution of the cloud system, airborne raindrop spectra and ground rainfall within 5 km from the operation centre under the action of acoustic waves. Therefore, a comprehensive meteorological monitoring system was constructed to examine the acoustic interference effect of the atmosphere and investigate the physical characteristics of clouds and rain droplets before and after acoustic operation. The present work uses a series of monitoring devices to investigate the dynamic processes impacting meteorological conditions, macroscopic responses of cloud-precipitation systems, and near-surface distribution of rainfall during acoustic operations.

The outline of this study is as follows. Section 2 introduces the field site and devices used for acoustic interference. Multisource monitoring data with different degrees of spatial and temporal precision obtained from in situ experiments were introduced. The experimental procedure and data analytical method are described in Section 3. A consistency verification of the monitoring devices is presented in Section 4 along with discussions of acoustic interference effects on meteorological systems, microphysical structures of aerial and near-surface droplets, microwave parameters, and distributions of near-ground rainfall during field operation. Finally, the findings are summarized in Section 5.

## 2. Experimental Set-Up

### 2.1. Experimental Site

The field site for acoustic interference testing was adjacent to the S101 provincial road, 20 km east of Darlag, in the southern part of the Golog Tibetan Autonomous Prefecture in China. The centre coordinates of the operation base are 33.55° E and 99.95° N with an altitude of 4098 m above sea level. Darlag is in the south-eastern region of the Qinghai-Tibet Plateau and has a plateau continental climate, which is significantly affected by the south-west monsoon during the summer. The region is characterized by low temperatures, large temperature differences between day and night, and low rainfall. The annual precipitation is mainly concentrated between May and September, which is the most suitable period for rainfall enhancement tests. The Jimai River, Darlag River, and Koqu River in Darlag are all fed from the main stream of the Yellow River. The experimental site is in the centre of the Jimai River Basin in the source region of the Yellow River (Hereafter referred to as SRYR, Figure 1a). The Jimai River is 101 km long, with a basin control area of 1852 km$^2$ and an average annual runoff of 451 million m$^3$.

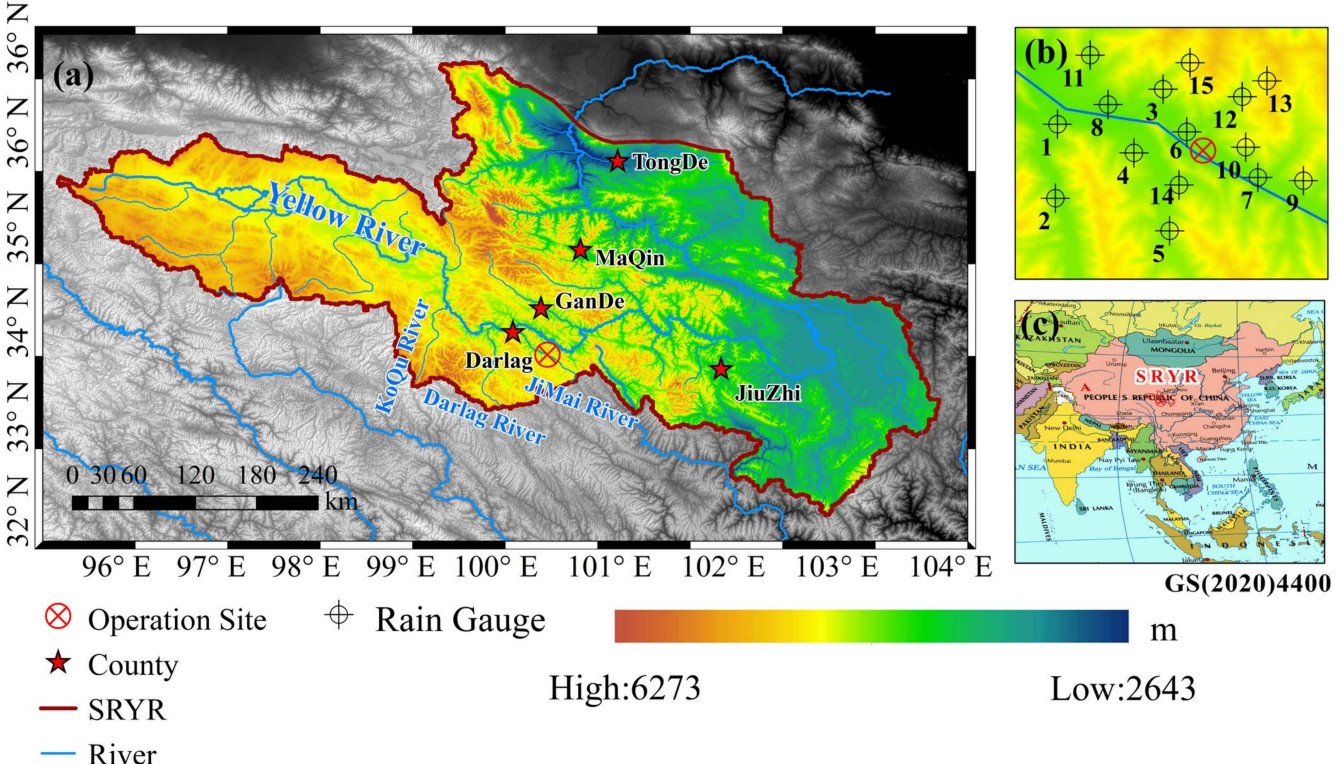

**Figure 1.** Geographic location of acoustic interference experiments in the SRYR (**a**) and distribution of rain gauges near the field site (**b**), and the location of SRYR in the world (**c**).

The sound generator and monitoring devices at the field site were arranged in a zigzag pattern, covering an experimental area of approximately 300 m$^2$. The site has four sound generators, a Ku-Ka dual-band weather radar instrument, a wind profile radar (WPR) instrument, a microwave radiometer (MWR), a micro-rain radar (MRR) instrument, a white weather station, three disdrometers (OTT), and 15 rain gauges (RGs, Figure 1c). The locations of these rain gauges (within a 5 km radius of the operation site) are presented in Figure 1. The observation system was constructed to monitor weather conditions, cloud motions, raindrop microphysical structures, and rainfall distributions. There are several Global Navigation Satellite System (GNSS) receiving stations near the operation area for local, real-time, high-resolution monitoring of airborne water resources.

## 2.2. Experimental Devices

The acoustic emission system comprised an acoustic generator, acoustic radiator, control system, power supply, and rotational station [77]. The acoustic generators emitted low-frequency (20–250 Hz), high-intensity (up to 156 dB at 1.0 m from the horn exit) acoustic waves toward the target cloud to promote collisions and the agglomeration of droplet aerosols into precipitation. The acoustic radiator was used to gather acoustic energy and to protect operators from exposure to scattered acoustic fields. An integrated monitoring system was constructed to monitor the occurrence, development, and extinction of cloud-precipitation systems and to obtain physical evidence that could be used to evaluate the effects of acoustic interference. The following aspects were monitored using the applied system: (i) the weather conditions during the experiment (e.g., ambient temperature, humidity, air pressure, wind field); (ii) the macroscopic characteristics of clouds (e.g., altitudes of cloud base and top, cloud thickness, concentration of condensation nuclei, liquid water and water vapour contents); (iii) the microphysical structures of droplets and near-surface precipitation (e.g., local concentration, drop spectrum distribution [DSD], morphological characteristics); and (iv) the microwave meteorological parameters (e.g., radar detection of cloud echo intensity, Doppler velocity and spectral width, temporal and spatial distributions of rainfall; rainfall rate near the ground).

The detection limitations and inversion method of each instrument make it difficult to invert the water vapour and cloud-precipitation systems around the field site by adopting a single approach. Therefore, multi-source monitoring data with different degrees of spatial and temporal precision were compared, validated, and integrated. On a time scale, the monitoring equipment can observe the entire cloud formation and precipitation processes, from development to decay. On a spatial scale, the environmental and weather monitoring capabilities varied among devices, although the overall spatial range covered an area on the order of 1 to $10^6$ m². In particular, the Ka−Ku dual-band weather radar instrument considers the advantages of the Ka band (35.5 GHz $\pm$ 25 MHz) and Ku band (13.6 GHz $\pm$ 25 MHz) for the meteorological measurements of clouds and rain, respectively. The distribution, echo intensity, radial velocity, velocity spectral width, and dual polarization parameters of microdroplets in air can therefore be accurately detected. Rain gauges and raindrop disdrometers monitor the microphysical structures of rainfall near the ground, including the raindrop distribution, raindrop concentration, rain rate, and accumulated rainfall. Meanwhile, microwave radiometers (K-band, 22–30 GHz; V-band, 51–59 GHz), WPR devices (L-band, 1270–1295 MHz), and weather stations monitor the meteorological environment. Specifically, the MWR detects the vertical profiles of temperature, relative humidity, water vapour density, and liquid water content within 10,000 m of altitude. The WPR uses the scattering effect of atmospheric turbulence on electromagnetic waves to detect the vertical profile and direction of wind speed, thus allowing real-time detection of three-dimensional wind fields, which provides dynamic evidence of large-scale rainfall cloud drifting motions.

## 3. Data and Methodology

### 3.1. Test Procedure

For the field operations, meteorological satellites (e.g., the Fengyun series) and a series of ground-based radar devices were used to monitor cloud echoes and forecast water vapour movements within 10 km of the site. Meteorological stations, WPR, and MWR were used for environmental detection to analyse the water vapour field and turbulence field near the experimental area, thereby enabling dynamic analysis and real-time forecasting of precipitation conditions. Once a cloud over the operating area meets the precipitation conditions, i.e., a reflectivity factor of >30 dBZ, random tests are carried out to determine whether an acoustic interference test will be performed:

(i) if the acoustic operation is deemed suitable, the acoustic generator is continuously opened for 40 min, and comparative observations (before and after the acoustic operation)

are conducted for 40 min. Thus, a total of 120 min was allotted for a test undergoing acoustic interference.

(ii) if the random experiment indicates no acoustic operation, a non-intervention control sample is collected using multi-source monitoring devices with a sampling period of 120 min. Among them, the natural monitoring period is 40 min, followed by acoustic intervention on the target clouds for 40 min until the sound generator is stopped, and the monitoring devices are continuously employed to observe variations of macro- and micro-characteristics of precipitation cloud systems after the withdrawal of acoustic waves.

Weather radar instruments and an MWR were used to monitor variations in the water vapour content and phase state of clouds, while disdrometers and rain gauges were employed to monitor the near-ground distribution of rainfall and droplet spectra within 5 km of the operation site. Microdroplet and meteorological parameters detected during the experiment were compared to available physical data. The direct and indirect effects of acoustic operations on cloud-precipitation systems were evaluated in this experiment. The direct effects refer to whether the sound wave impacts the cloud distribution and cloud droplet microphysical structure (e.g., cloud thickness, cloud base height, ice crystal and cloud droplet concentrations); the indirect effects refer to the influence of the sound waves on precipitation conversion (e.g., rain rate, precipitation area, cumulative rainfall).

### 3.2. Data Processing

The resolution and accuracy of cloud and rain characteristics and meteorological parameters differ among monitoring devices, and therefore, analysis of operational effects should verify the quality of observations. In this work, outliers or noisy datapoints in the rainfall samples were removed, i.e., the monitoring series with R < 0.01 mm/h or a total number of raindrops <10. In addition, the meteorological and precipitation parameters derived from different devices typically overlap, while the observation principle varies depending on the device; this requires consistency verification for all monitoring samples. Device consistency can be evaluated by observing the same physical quantities using different devices, which would reflect trends and differences in physical quantities. It is also necessary to consider the distinct spatial and temporal characteristics of each monitoring series and integrate them into high-quality samples. The calibration and assimilation of different monitoring samples can enable mutual verification at the same sampling instant and location. For a series of meteorological radar devices, it is necessary to conduct internal and external calibrations of the reflectivity factor and radar polarization parameters measured by each device. The consistency of a device can be determined by the data error, correlation, and skewness. The mean difference (MD) and normalized standard error (NSE) are expressed as shown in Equations (1) and (2), respectively,

$$MD = \frac{1}{N} \sum_{i=1}^{N} (Xi - Yi) \tag{1}$$

$$NSE = \frac{\frac{1}{N} \sum_{n=1}^{N} (Xi - Y_i)}{\frac{1}{N} \sum_{n=1}^{N} Y_i} \tag{2}$$

where $N$ is the sampling number, and $X$ and $Y$ represent samples from different devices.

The correlation can be expressed using the Pearson correlation coefficient (CC), which is computed using Equation (3),

$$CC = \frac{\sum_{i=1}^{N} (Xi - \overline{X})(Yi - \overline{Y})}{\sqrt{\sum_{i=1}^{N} (Xi - \overline{X})^2} \sqrt{\sum_{i=1}^{N} (Yi - \overline{Y})^2}} \tag{3}$$

where $\overline{X}$ and $\overline{Y}$ represent the average values of the $X$ and $Y$ samples, respectively. Here, $CC$ values of 1, $-1$, and 0 indicate completely positive correlation, completely negative correlation, and no correlation, respectively.

## 4. Results and Discussion

The temporal and spatial variations in meteorological conditions, the precipitation cloud system, droplet microphysical structures, and near-surface rainfall distribution were analysed during acoustic operations based on the multi-dimensional and multi-scale monitoring system to provide physical evidentiary support for the interference effects of low-frequency acoustic waves. In this work, we evaluated the consistency of each monitoring instrument in terms of environmental and precipitation measurements (Figure 2).

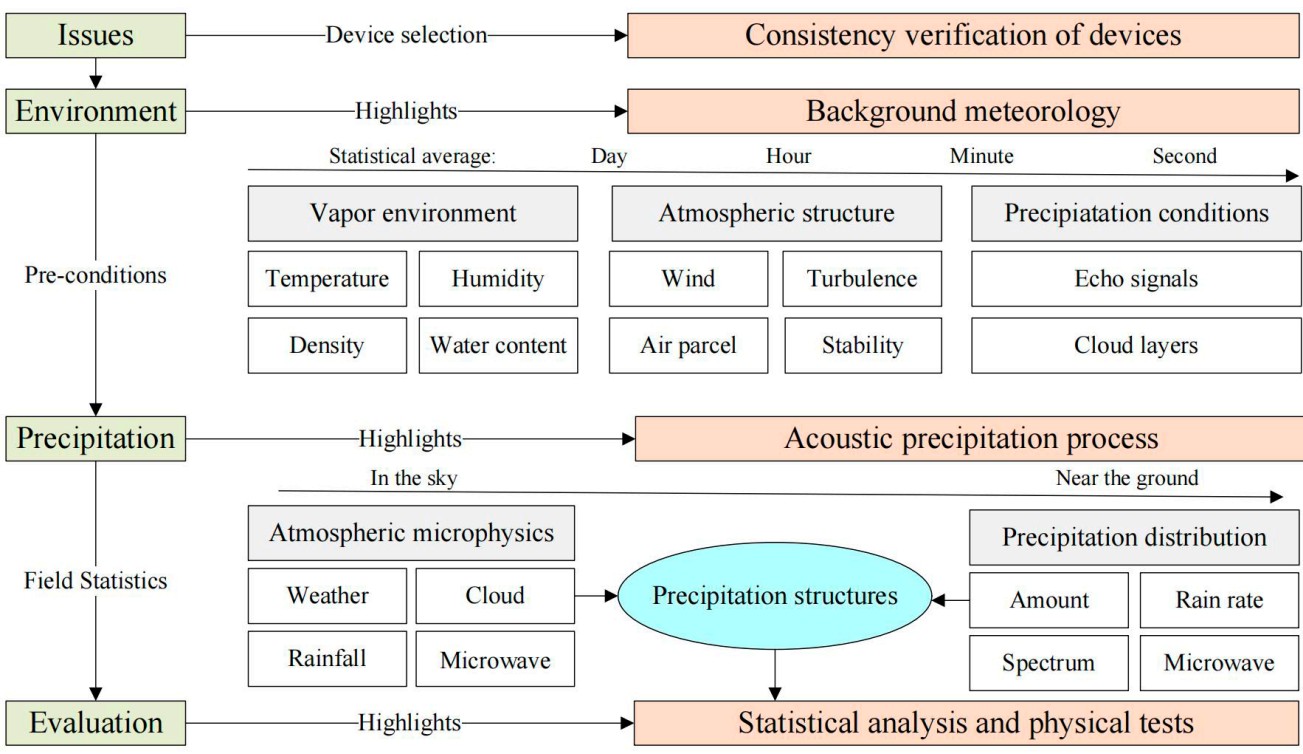

**Figure 2.** Meteorological measurements and effectiveness evaluation processes.

### 4.1. Device Consistency Verification

The detection parameters of different monitoring devices have a certain degree of overlap. The monitoring sample on 17 September 2019 was used for consistency analysis. Figure 3 compares the rain rates collected by MRR and OTT, which had *MD* and *NSE* values of 0.25 and 0.34 mm/h, respectively. The CC index relating the *R* values obtained by MRR at 100, 200, and 330 m from the ground and those measured by the surface raindrop disdrometers at the field site were 0.79, 0.83, and 0.80, respectively, indicating a high correlation between MRR and OTT below a sampling height of 300 m. In terms of meteorological parameters, the *MD* (and *NSE*) values of the reflectivity factors (*Z*) collected by the Ka−Ku dual-band radar and the MRR at 600, 1200, and 1800 m from the ground were 1.48 (0.10), 0.67 (0.03), and 0.33 (0.02) dBZ, respectively. The Pearson correlation coefficients relating the *Z* values acquired by the Ka−Ku weather radar in the Ka band at 600, 1200, and 1800 m from the ground and the corresponding MRR-derived *Z* values near the ground were 0.93, 0.87, and 0.78, respectively. These results indicate a strong correlation (i.e., >0.7) between the two radar methods using similar detection wavelengths.

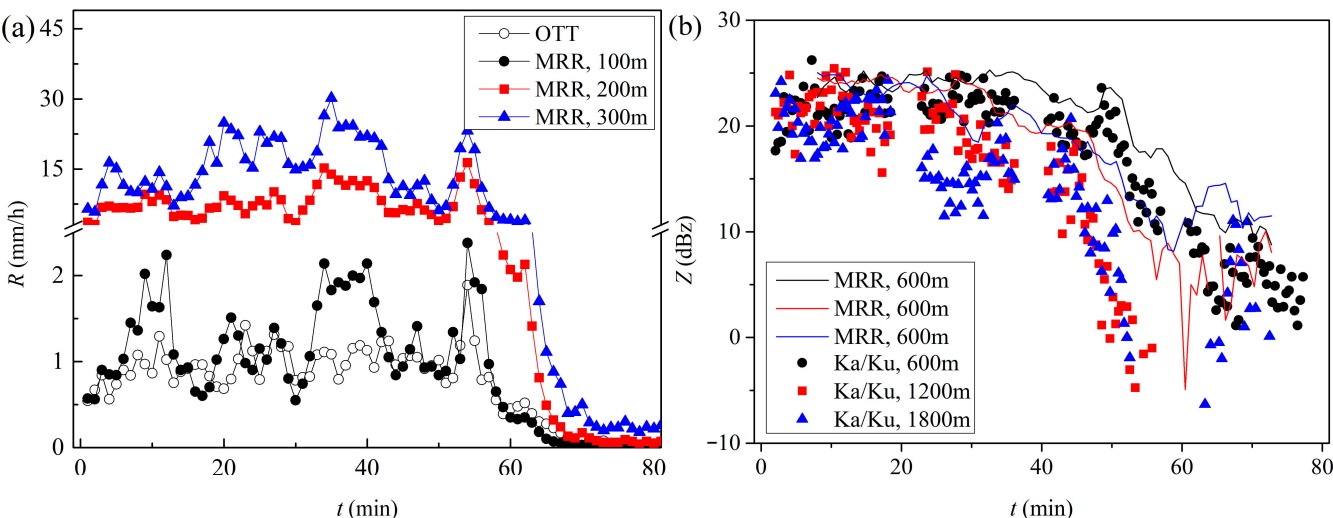

**Figure 3.** Consistency verification of monitoring devices. (**a**) *R* values obtained by MRR and OTT; (**b**) *Z* values obtained by Ka−Ku and MRR.

### 4.2. Meteorological Background near the Site

　　An acoustic interference process with a stratiform system in Qinghai Province during the summer monsoon period (17 September 2019) was used to evaluate the multi-source monitoring results. Figure 4 shows the meteorological conditions near the field site at an altitude of 500 hPa at 6:00. The generalized potential temperature (GPT) characterizes the distribution and variation in temperature and humidity in the real atmosphere. Figure 1a shows that the field site (in the Darlag region) was located at the eastern edge of the warm centre before precipitation occurred with a GPT magnitude of 347 K. The weather-scale cyclonic vortex formed and moved eastwards in the eastern part of the Tibetan Plateau, accompanied by the development of high temperature and humidity. There was strong wind shear in the western part of the operation area before precipitation occurred. This implies that there was sufficient precipitation potential. Moreover, the operation site was more enriched in water vapour compared to the surrounding inland areas, with a magnitude of approximately 12 g/s·hPa·cm, thereby providing better conditions for acoustic interference (6:40–8:00).

　　Atmospheric turbulence and wind dynamics can significantly influence the evolution of clouds and precipitation. Figure 5a shows the wind speed structure over the field site during acoustic operation. The southeast wind is at an altitude of 3000–6000 m above the field site, and the horizontal wind speed reaches 30 m/s. The westerly wind is distributed over an altitude of 1500–3000 m, with the horizontal wind speed reaching 10 m/s. The sampling interval of 500–1500 m is dominated by northwest winds, with a wind speed up to 4 m/s. There is a clear wind shear line at an altitude of approximately 2700 m, and the ambient wind speed is significantly shifted. When the acoustic operation was finished (at 8:00), the vertical wind speed at sampling heights between 0 and 400 m decreased. Figure 5b shows the variations in the atmospheric refractive index ($cn^2$) during the operation time. The $cn^2$ value describes the strength of the atmospheric turbulence, which characterizes the random inhomogeneity of the atmospheric refractive index. When there were clouds or precipitation, the strength of the echo signal received by the WPR increased significantly, and therefore, the $cn^2$ derived from the signal-to-noise ratio also increased. During the opening period of the acoustic generator (6:40–7:20), the $cn^2$ value was relatively high at altitudes less than 4000 m above sea level, indicating a high possibility of precipitation. During the sampling period when there was no acoustic interference (7:20–8:00), the $cn^2$ index exhibited a decreasing trend in the zenith direction of the experimental area, indicating that the rainfall probability was reduced.

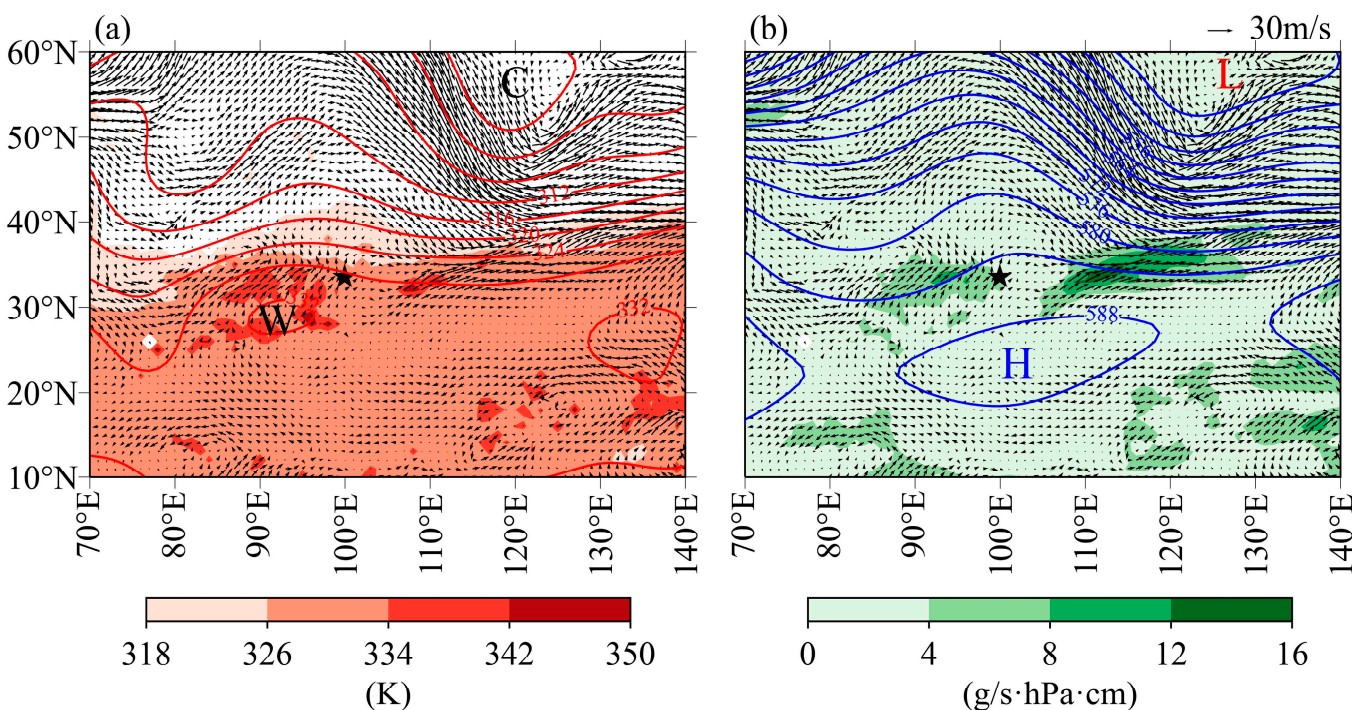

**Figure 4.** Meteorological conditions near the field site at an altitude of 500 hPa, where the pentagram symbol indicates the operation centre. The red and green filling blocks are the warm zones (**a**) and water vapour fluxes (**b**), respectively. The red and blue solid lines are the generalized potential temperature and potential heights, respectively. The capital letters H, L, W and C represent the locations of the high-pressure centre, low-pressure centre, warm centre and cold centre, respectively. The arrow represents the wind vector. (Final Operational Global Analysis data from American National Centre for Environment Prediction; http://rda.ucar.edu/datasets/ds083.2, accessed on 11 November 2020).

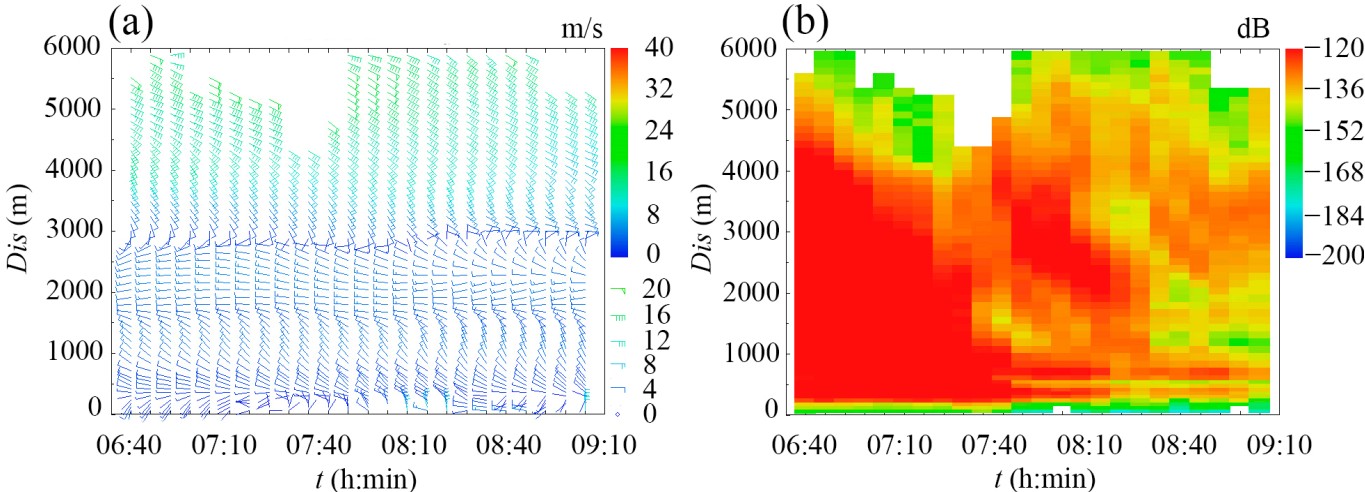

**Figure 5.** Temporal evolution of (**a**) wind speed and (**b**) atmospheric refractive index ($cn^2$) measured by WPR in the zenith direction of the field site.

### 4.3. Variations in the Precipitation Cloud System

Figure 6 shows the reflectivity factor ($Z$), radial velocity ($V_r$), and velocity spectral width ($W$) near the field site obtained by weather radar instruments after acoustic operation. During the initial stage of acoustic operation (Figure 6a), there is a stable brightness band at a detection distance of 600–800 m, i.e., the altitude of the zero-degree band is in the range

of 223–297 m according to a pitch angle of 21.8°. During the acoustic operation period (Figure 6b,c), there is a significant surge in the $Z$ value within 1200 m of the operation centre; specifically, the average $Z$ value increases from 25.87 to 28.58 dBZ in the range of 100–500 m. When the acoustic interference stops (Figure 6c), the spatial inhomogeneity of the echo intensity of the surrounding cloud becomes clear. That is, the echo intensity is higher at distances as far as 3 km only in the azimuth ($A_z$) of 100–240°, whereas echoes in the rest of the azimuths are weaker, with an average value of 16.19 dBZ. When the acoustic wave stopped for 40 min (Figure 6e), the $Z$ values of the clouds around the operation centre gradually decreased to 4.69 dBZ with a somewhat uniform distribution. This trigger effect of sound initiation and termination has also been found in the work of Wei et al. [6] and Qiu et al. [47].

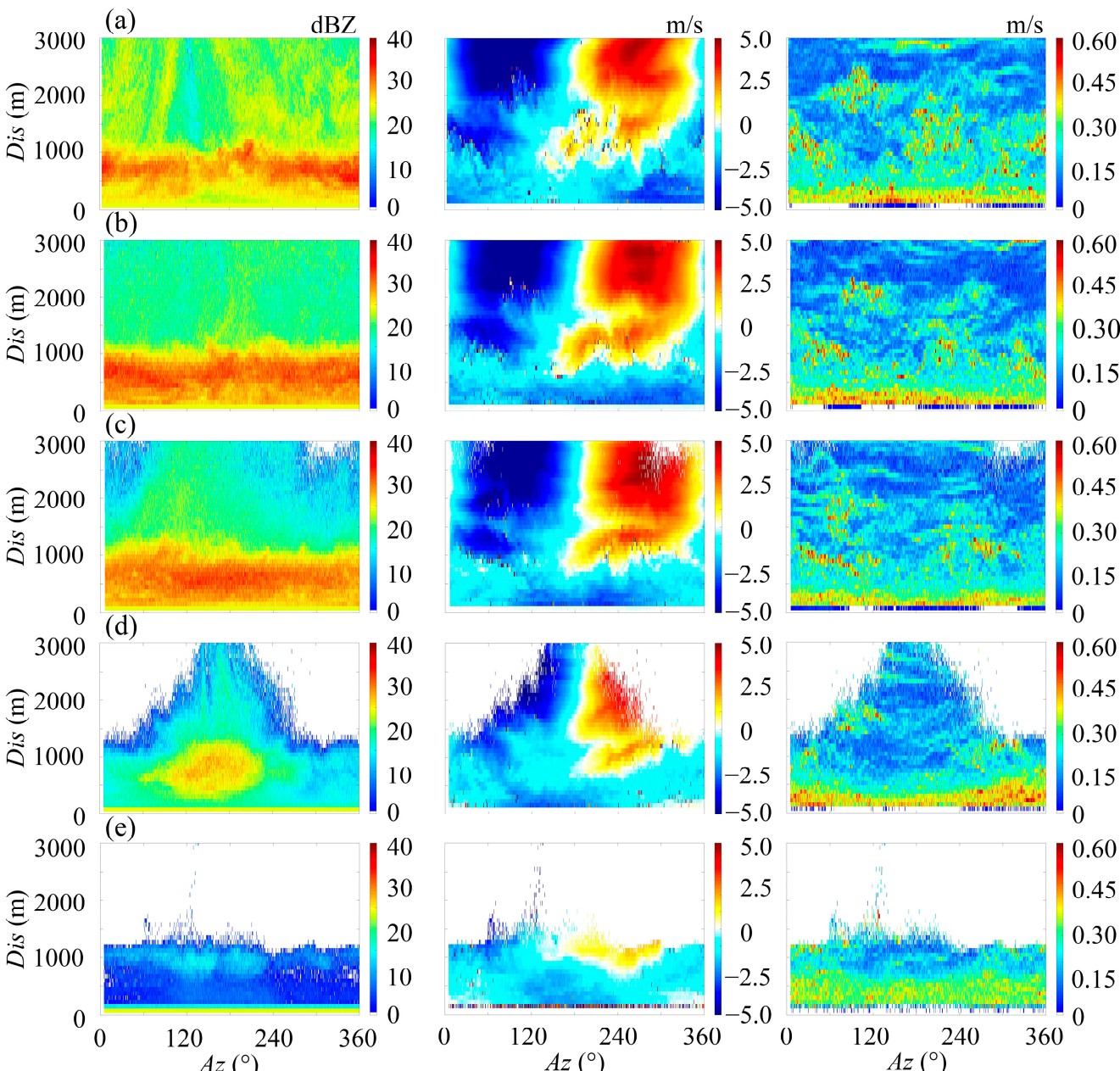

**Figure 6.** Temporal evolution of $Z$ measured by cloud and weather radar instruments in the Ka band (Plan Position Indicator, or PPI mode, pitch angle = 21.8°). The angle interval covered by the dashed window is the radar azimuth angle in the Range Height Indicator, or RHI mode. (**a**) $t_1$ = 6:40, (**b**) $t_2$ = 7:00, (**c**) $t_3$ = 7:20, (**d**) $t_4$ = 7:40, (**e**) $t_5$ = 8:00. The legend represents $Z$, $V_r$ and $W$ from left to right.

The droplet $V_r$ around the operation centre is stable, and the average value within 600 m of the operation centre is negative. The average value of $V_r$ with the acoustic intervention is $-1.62$ m/s, which changes to $-1.32$ m/s after 40 min of acoustic operation. This result reflects the high kinetic energy of microdroplets during acoustic interference. When the sampling distance is greater than 600 m, the $V_r$ distribution is stable, i.e., $V_r$ is negative for $A_z$ of 210°–340° and positive for $A_z$ of 30°–150°. This indicates that the air flow field around the working area is relatively stable. The Doppler velocity spectral width characterizes the deviation of the microdroplet Doppler velocity from its mean value; such deviations are caused by the various radial velocities of scattered droplets. The microdroplet motion is influenced by the mean wind field, gravity, and ambient turbulence. Therefore, the main factors impacting fluctuations in $W$ include the vertical wind shear, atmospheric turbulent motion, and inhomogeneous falling velocities of droplets. Figure 5 indicates that there is no significant vertical wind shear at sampling heights below 1100 m around the operating site. This implies that the large spectral width fluctuations in Figure 6a–c are likely due to oscillations of fluid media and raindrops caused by acoustic interference. The DSD directly determines the detected $W$ value. The standard deviation (*STD*) of $W$ without acoustic interference within 500 m of the operation centre is 0.044 m/s, which is an average of 0.028 m/s lower than the corresponding value measured during acoustic operation. In the region 800 m away from the operation site, distinct patches of localized areas with high $W$ values emerge during the operation period (with *STDs* and sample ranges of 0.096 and 0.86 m/s, respectively). In contrast, there is very little area with high $W$ in the non-operational or monitoring periods without acoustic interference.

Figure 7 presents the cloud bottom height ($H_c$), integrated water vapour density ($I_v$), and integrated liquid water content ($I_l$) over the operation site. Within 50 min after the acoustic interference, the $H_c$ value was stable and less than 0.3 km in the zenith, south, and north directions. When the acoustic device was turned off for 10 min, the $H_c$ value rapidly increased from 0.3 to 2.2 km and then remained stable. The response time of the trigger effect of the acoustic wave is consistent with the work of Qiu et al. [47]. The values of $I_v$ and $I_l$ in the zenith direction during acoustic operation were 2.5 cm and 2.0 mm, respectively, which were significantly higher than the corresponding values in the southwards and northwards directions. This was attributed to the uneven distribution of water vapour or acoustic effects. In addition, there was no significant change in $I_v$ over the field site during the acoustic operation. However, the $I_l$ values under acoustic interference were slightly larger than the corresponding values that were not subjected to acoustic waves.

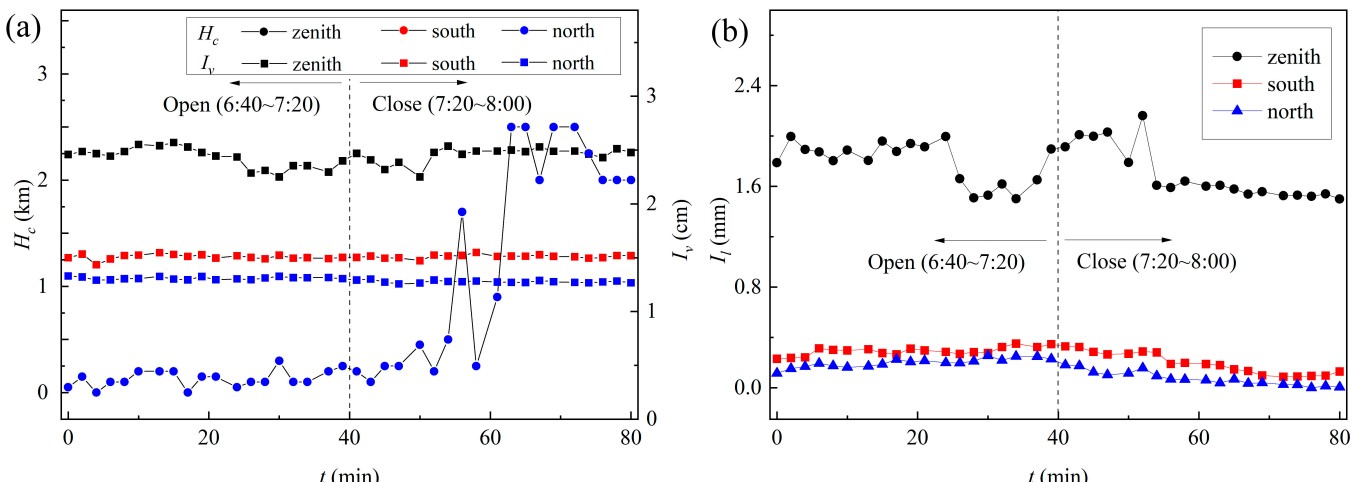

**Figure 7.** Temporal evolution of $H_c$, $I_v$ (**a**), and $I_l$ (**b**) measured by MWR in the zenith direction of the field site.

### 4.4. Vertical Structure of Liquid Droplets

The vertical profiles of *R* and *Z* over the zenith of the operation centre during the acoustic operation are presented in Figure 8. The monitoring samples were decomposed into four segments (i.e., 20 min intervals), and the *Z* and *R* values of each sequence were time-averaged along the zenith direction. The *R* value gradually increased with decreasing altitude within the sampling height ranges of 300–400 m and 1800–2400 m, and there were gradient bands of rain rates in the range of 100–400 m above sea level. The average *R* value within 1800–2400 m during 20–40 min of acoustic operation was significantly lower than that during the first 0–20 min, i.e., the beginning of the operation. At altitudes of 500–3100 m, the lower the sampling height is, the higher the *R* value between 7:00 and 7:20. Within 20 min after stopping acoustic interference (i.e., 7:20–7:40), the *R* value above 1800 m decreased significantly and was only larger (by ~10 mm/h) within an altitude of 100–400 m. In terms of the reflectivity factor under acoustic operation (6:40–7:20), *Z* decreased as the elevation increased from 400 to 3100 m, and a large gradient band of *Z* values appeared near the ground (from 0 to 400 m). After the acoustic generator was shut down (7:20–8:00), the *STD* of *Z* significantly decreased. The *Z* value is positively correlated with the sampling height when the altitude is >2400 m, which is consistent with the corresponding *R* values.

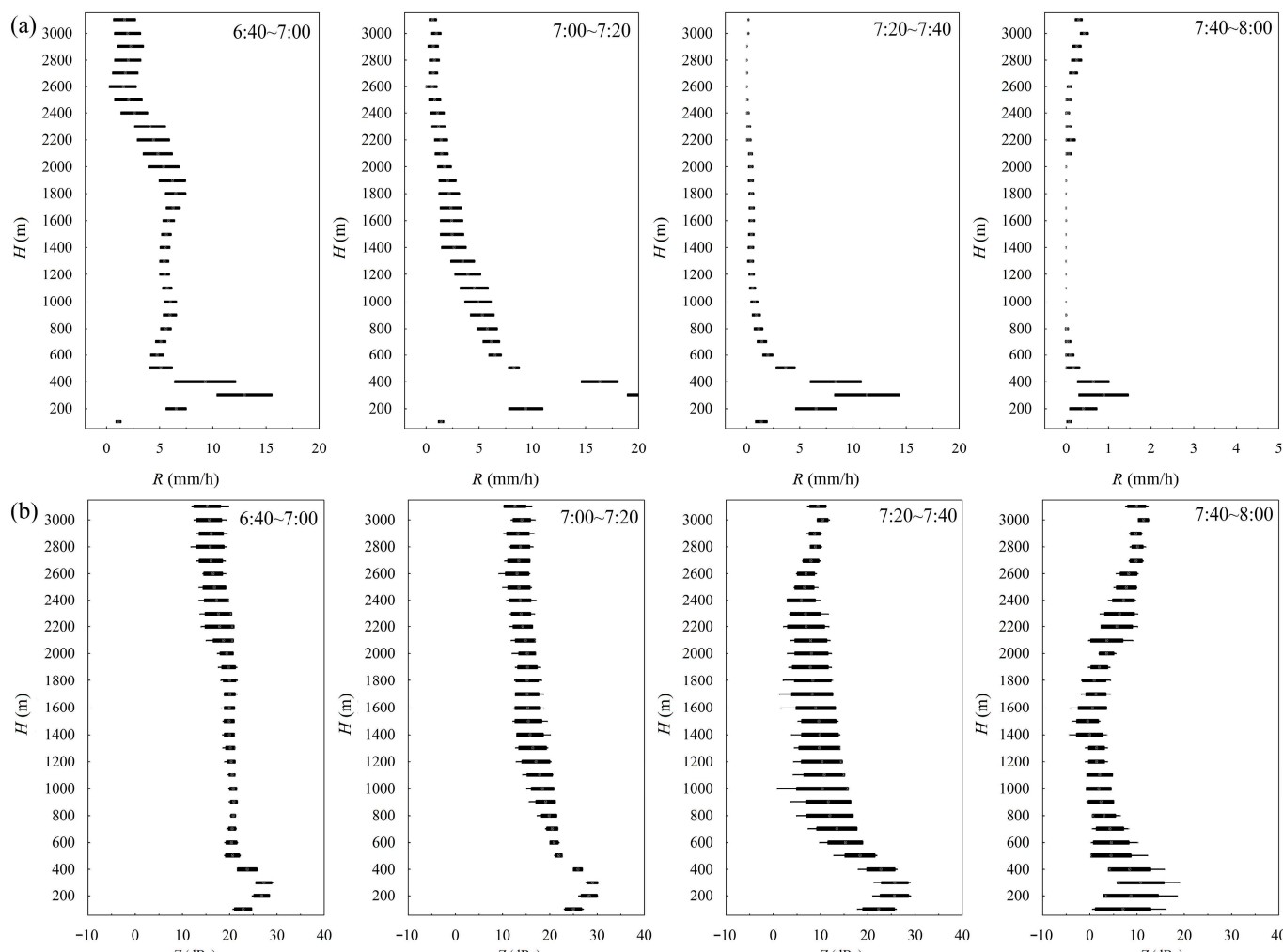

**Figure 8.** Temporal evolution of (**a**) *R* and (**b**) *Z* profiles measured by MRR in the zenith direction of the field site.

Figure 9 presents the temporal evolution of droplet concentrations and the corresponding variations in the zenith direction of the site under the action of acoustic waves. In general, the particle size in the zenith direction of the site during acoustic operation was in the range of 0.3–4.2 mm. For sampling heights >200 m, the concentration of droplets with diameters ($d$) < 1 mm is dominant. The concentration of particles with $d > 3$ mm is lower at altitudes of 400–1500 m. The high-power acoustic wave did not significantly change the overall distribution of the raindrop concentration profile, although it did affect the particle distribution in the local region. For example, the concentrations of large particles ($d > 4$ mm) and relatively small particles ($d < 1.5$ mm) were reduced after the acoustic operation. Figure 9a shows the difference in droplet concentrations in the zenith direction of the experimental site with 20 min of acoustic interference (6:40–7:00) versus natural detection (6:00–6:40). In general, the particle concentration increased significantly by 1–3 ln [$1/m^3\cdot$mm] at all size scales in certain areas below 400 m or above 2200 m that were subjected to acoustic waves. The increase or decrease in particle concentration in the sampling height range of 400–2200 m occurred randomly, which indicates that particle agglomeration and deagglomeration were in dynamic equilibrium. During the period of acoustic activation (Figure 9a,b), the concentration of particles with $d = 1$–1.5 mm in the sampling height range of 500–1700 m decreased significantly, which indicates a spatial and physical shift of the particles. When the acoustic operation stopped (Figure 9c,d), the particle concentration decreased in most areas above the operating region, especially from 0 to 500 m. This indicates a gradual end to the precipitation.

### 4.5. Horizontal Structure of Near-Ground Rainfall

Figure 10 shows the distribution of cumulative ground rainfall and rain rates in the vicinity of the site for 40 min before, during, and after acoustic interference. During the 40 min period before the acoustic interference (Figure 10a), the cumulative rainfall on the ground was generally greater in the south than in the northwest, and it was particularly high at monitoring points #2, #5, #7, and #12, with rainfalls of 0.6, 0.6, 0.6, and 0.5 mm, respectively. When the acoustic generator was operated for 40 min (Figure 10b), the accumulated precipitation was generally higher in the south and lower in the north, with particularly higher values at monitoring points #5 and #14 (~1.5 mm). The spatial distribution of accumulated rainfall did not vary significantly after stopping the acoustic action (Figure 10c), and the maximum accumulated rainfall was measured at monitoring point #5 (1.6 mm). The acoustic interference has a considerable effect on rainfall enhancement, which was consistent with the findings of Wang et al. [48]. In terms of the rain rate, the time-averaged $R$ value within 5 km of the operation centre was relatively uniform before acoustic operation (~0.7 mm/h). Compared with the natural control, the spatial distribution of $R$ changed significantly after acoustic interference, which was consistent with the distribution of accumulated precipitation. The $R$ values at measurement points #5 and #14 correspond to local peaks of 1.35 and 1.5 mm/h, respectively. Within 40 min after the acoustic generator was turned off, the $R$ value decreased significantly, and the average $R$ value within 5 km was 0.28 mm/h.

During the entire rainfall process, the $R$ value and accumulated rainfall were generally larger at measurement points #2, #5, and #7 in the south and #12 in the east. This may be because measurement point #12 is on the windward slope, where the local slope topography promotes high natural rainfall. In addition, horizontal winds over the operation area are in the northwest direction within the altitude range of 500–1500 m, with wind speeds up to 4 m/s. In contrast, the altitudes of 1500–3000 m are dominated by westwards winds, with horizontal speeds reaching 10 m/s. The height of the cloud base determined by MWR is in the range of 300–2200 m. This suggests that the precipitation clouds may shift toward the east or southeast under the influence of horizontal winds, resulting in high localized precipitation in those regions.

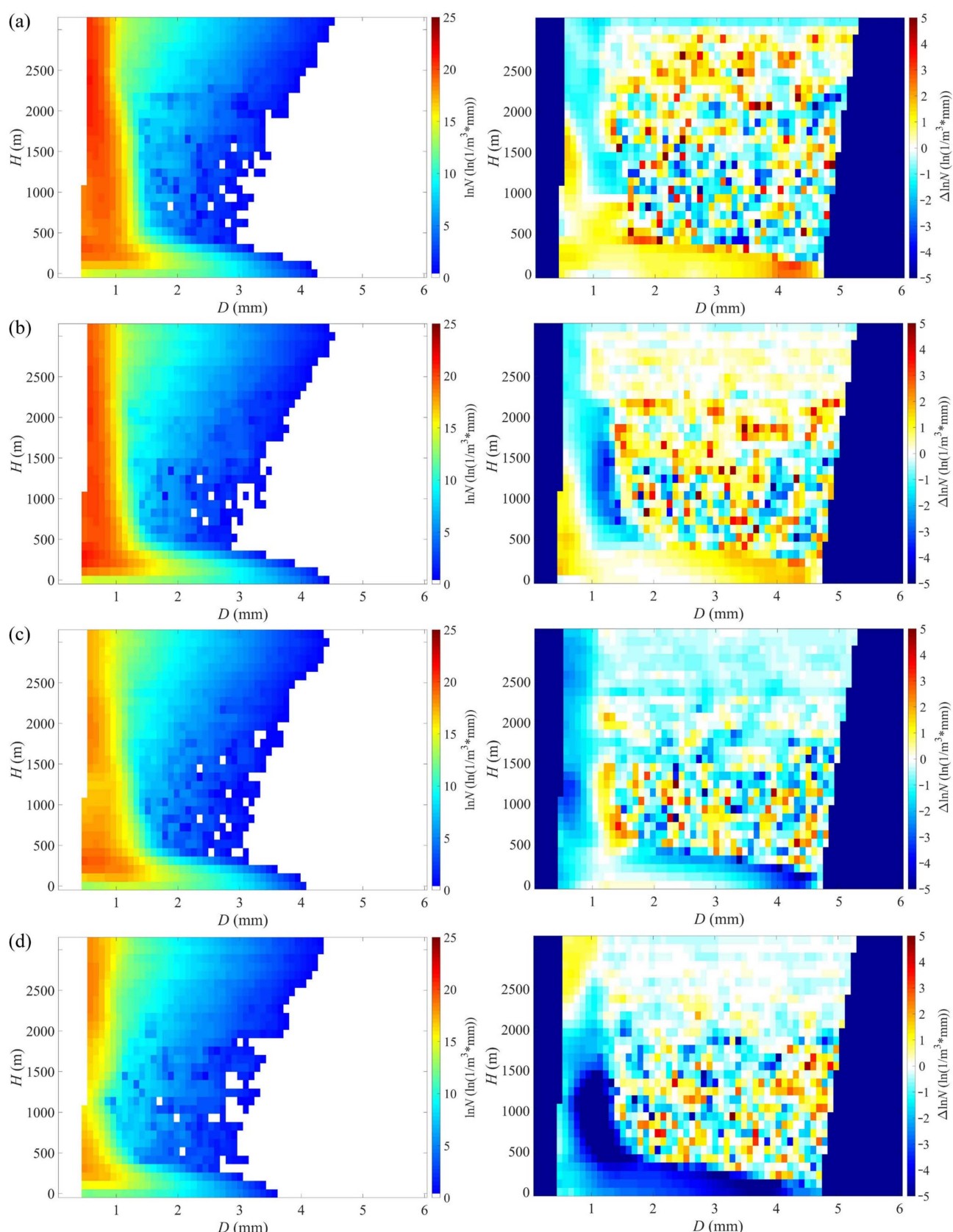

**Figure 9.** Temporal evolution of raindrop concentrations and the corresponding variations measured by MRR in the zenith direction of the field site. (**a**) $t_1$ = 6:40–7:00, (**b**) $t_2$ = 7:00–7:20, (**c**) $t_3$ = 7:20–7:40, (**d**) $t_4$ = 7:40–8:00.

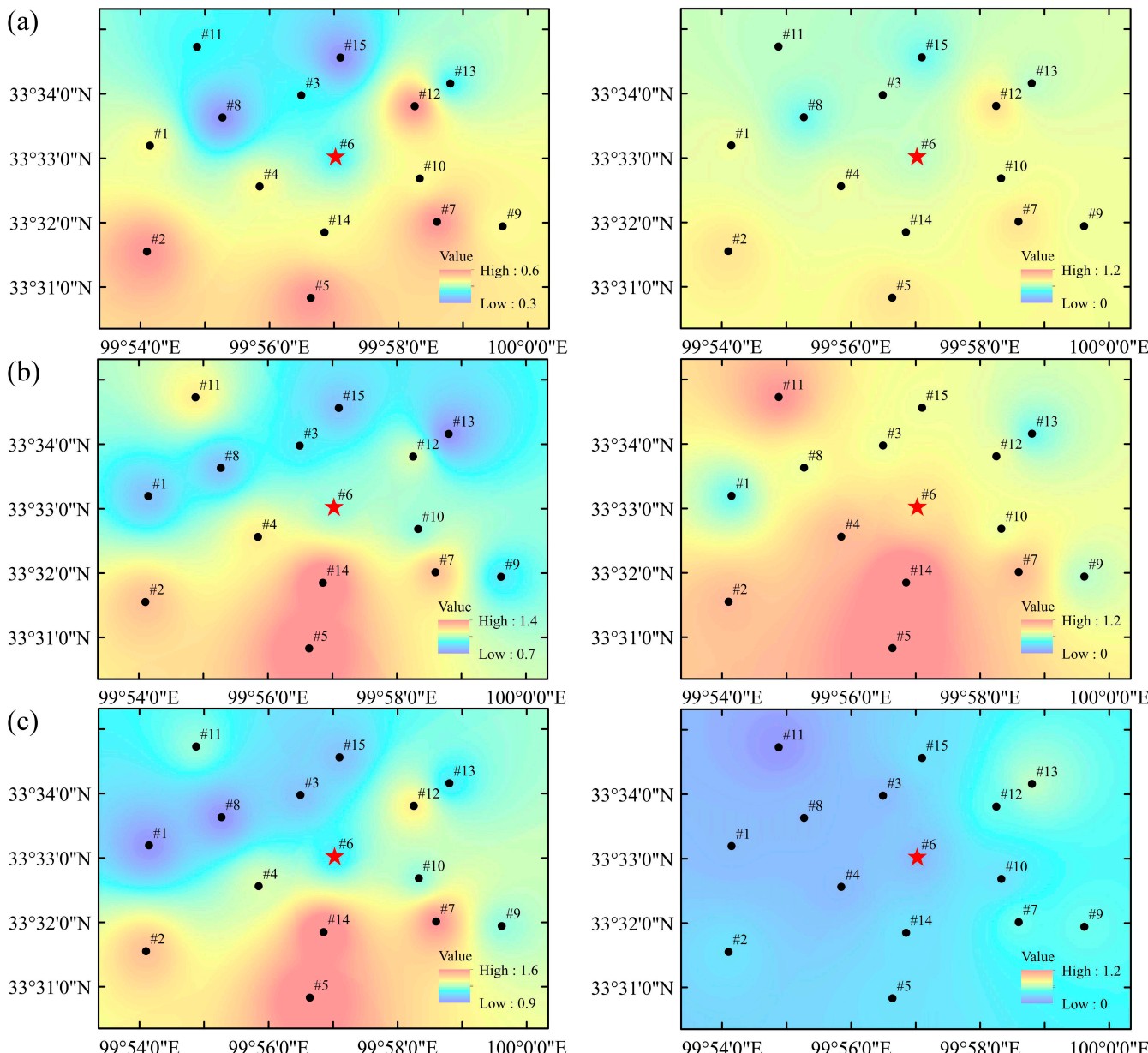

**Figure 10.** Temporal evolution of *R* and accumulated rainfall measured by rain gauges in the zenith direction of the field site. (**a**) $t_1$ = 6:00–6:40, (**b**) $t_2$ = 6:40–7:20, (**c**) $t_3$ = 7:20–8:00. The pentagram and solid circle symbols present position of field site and rain gauges, respectively.

The temporal variations in raindrop profiles collected by OTT near the ground at the operation centre are presented in Figure 11. The surface raindrop spectral width reached 10.3 mm, and the average spectral width of the droplet particle size was 7.5 mm. The concentration of droplets in the size range of 1.625 to 2.75 mm was relatively high under acoustic interference (40–80 min). However, after the acoustic equipment was turned off (40–80 min), the concentration in the size range of 3.25–4.75 mm increased. This result indicates that agglomeration and deconglomeration behaviours occurred during acoustic operation. In addition, Figure 11c,d shows that the raindrop falling velocity has a parabolic relationship with the particle size. In general, the larger the particle size is, the greater the droplet velocity. Raindrops with diameters of ~7.5 mm fall with a velocity greater than 10 m/s. Compared with the control monitoring period, the concentration of particles in the size range of 2.13–4.75 mm decreased during acoustic operation (0–40 min), whereas the concentrations of particles smaller than 2.13 mm and larger than 4.75 mm increased. This

indicates that particles with sizes ranging from 2.13 to 4.75 mm undergo a transformation, with some breaking into smaller particles and others coalescing into larger droplets. When the acoustic equipment was turned off (40–80 min), the trend was reversed, i.e., both larger particles (*d* > 4.76 mm) and smaller particles (*d* <2.125 mm) were transformed into particles with *d* = 2.13–4.75 mm.

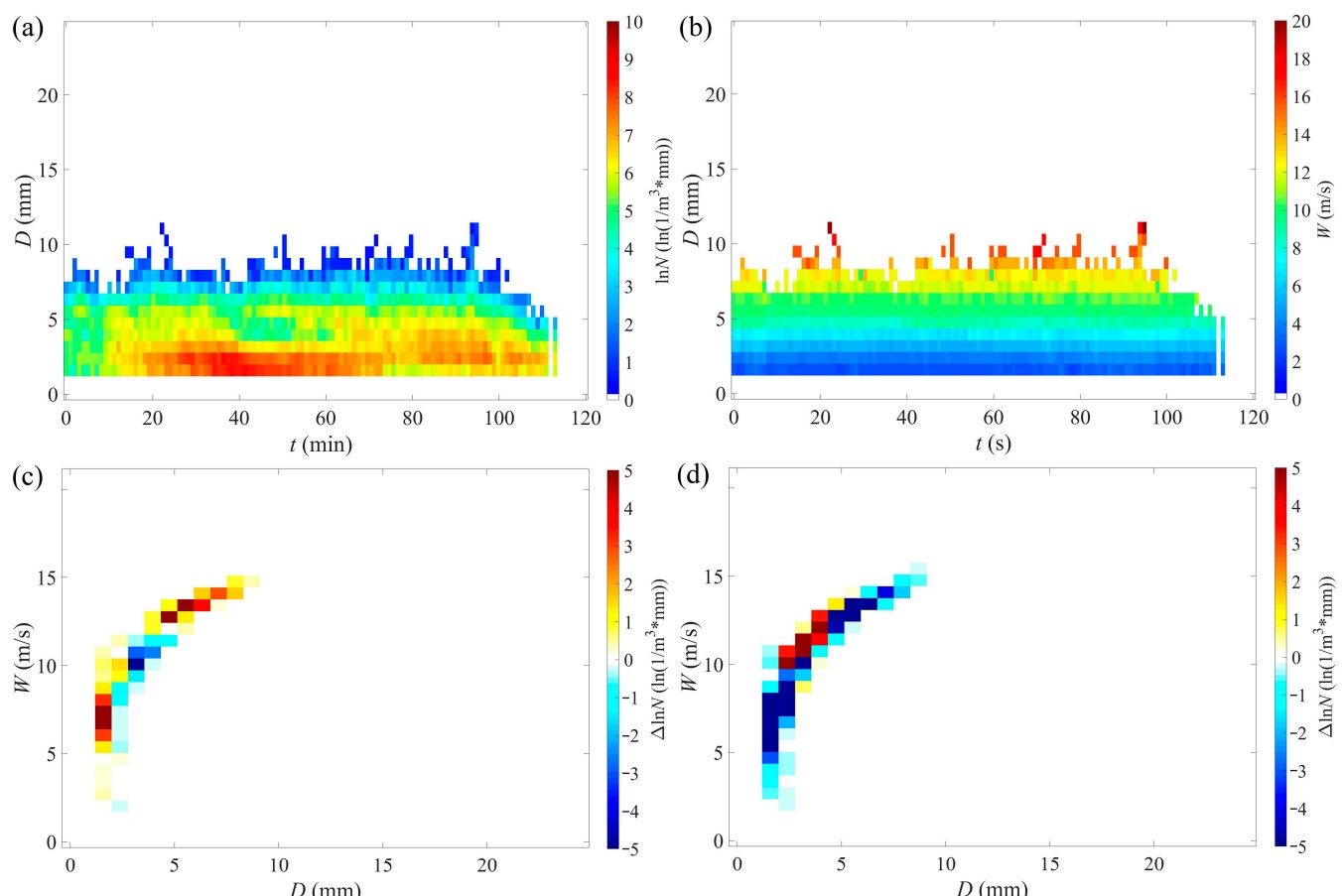

**Figure 11.** Temporal evolution of raindrop size distributions measured by OTT at the field site. (**a**) Concentration; (**b**) velocity; (**c**) *W–D* relationship under interference and natural conditions; (**d**) *W–D* relationship when turning the device on and off.

Variations in the *D*, *R*, *Z*, and particle kinetic energy ($e_k$) collected by OTT at the operation centre are presented in Figure 12. The grey shading in the figure represents the period of acoustic interference. The *D*, *R*, *Z*, and $e_k$ values fluctuated during the first 40 min of acoustic operation, where the overall *D* value gradually decreased to 0.6 mm, and the *R*, *Z* and $e_k$ values first increased and then decreased to 0.52 mm/h, 16.1 dBZ, and 3.4 kJ, respectively. When the acoustic wave interfered with the clouds, *D*, *R*, *Z*, and $e_k$ gradually increased to 0.92 mm, 1.24 mm/h, 22.5 dBZ, and 2.3 kJ, respectively. Upon stopping the acoustic interference, *D*, *R*, *Z*, and $e_k$ gradually decreased until the rainfall process stopped entirely (*t* = 114 min).

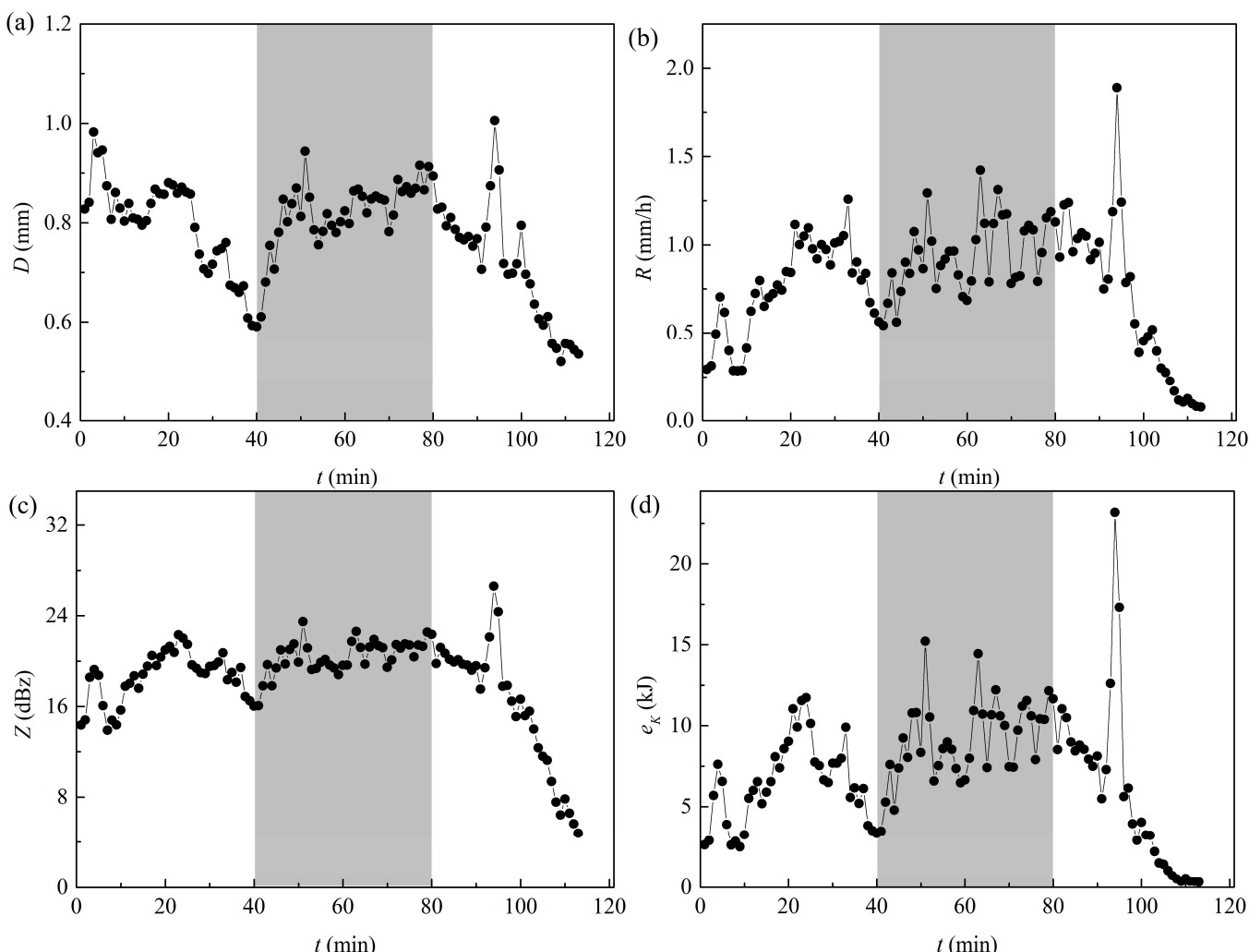

**Figure 12.** Temporal evolution of (**a**) *D*, (**b**) *R*, (**c**) *Z*, and (**d**) $e_k$ measured by OTT at the field site.

## 5. Conclusions

This study introduces cloud-precipitation interference technology based on acoustic agglomeration and describes key physical examinations. A multi-dimensional and multi-scale monitoring system was implemented to observe and evaluate the full lifecycle of stratiform precipitation in the SRYR during the East Asia summer monsoon period. The macro- and microphysical characteristics of clouds and precipitation were elucidated, and variations in the spatial and temporal distribution of ground rainfall were evaluated during acoustic interference. The main research findings are as follows.

(i) The evolution of clouds and precipitation were significantly affected by atmospheric turbulence and wind dynamics. The atmospheric refractive index could directly reflect the possibility of precipitation.

(ii) Acoustic intervention did not cause systematic changes in the precipitation cloud system, such as bright temperature structure, water vapour and liquid water content. However, the agglomeration and deconglomeration behaviours of clouds and rain droplets were in a dynamic equilibrium in the sampling altitude range of 400–2200 m under the action of acoustic waves. This was directly manifested as a trigger effect in the *Z* value and number concentrations of the droplet groups in the cloud when the acoustic intervention was turned on and off.

(iii) Near-surface precipitation and raindrop microphysical structures were directly influenced by acoustic operation with local rainfall accumulation. The dynamic transformation and agglomeration-deconglomeration equilibrium of raindrops (diameters of 2.13–4.75 mm)

was observed during acoustic interference and revocation. In addition, the spatial and temporal distributions of rainfall and rainfall intensity within 5 km of the field site were not uniform and were potentially influenced by the wind field and mountain slope.

Although low-frequency acoustic waves intervene in the evolution of cloud and rain systems to some extent, the physical characteristic response of precipitation in this study can only partially reflect the effect of acoustic intervention coupled with natural variability in the SRYR region during the East Asian summer monsoon. At this stage, it is difficult to fully disentangle the acoustic effects from the changing and evolving precipitation system unless more randomized comparison tests are available.

**Supplementary Materials:** The following supporting information can be downloaded at: https://www.mdpi.com/article/10.3390/rs15040993/s1, Table S1: comparison of cloud-precipitation interference operation.

**Author Contributions:** Conceptualization, J.W. and Y.S.; methodology, J.W.; validation, J.W., Y.S. and Z.Q.; formal analysis, Y.S. and Z.Q.; investigation, Y.S.; resources, J.W. and G.W.; data curation, J.W., Y.S. and Z.Q.; experimental design and conduct, J.W. and Z.Q., writing—original draft preparation, Y.S. and Z.Q.; writing—review and editing, J.W., Y.S. and Z.Q.; supervision, J.W. and G.W.; project administration, J.W.; funding acquisition, J.W. and Y.S. All authors have read and agreed to the published version of the manuscript.

**Funding:** This research was funded by the National Key Research and Development Program (Grant No. 2022YFC3202400, 2017YFC0403600); the National Natural Science Foundation of China (Grant No. 52009059, 52209092 and 91847302); the Joint Institute of Internet of Water and Digital Water Governance, Tsinghua-Ningxia-Yinchuan (SKL-IOW-2022TC2201); and the China postdoctoral science foundation (Grant No. 2021T140385, 2018M641372).

**Data Availability Statement:** The data presented in this study are available on request from the corresponding author.

**Acknowledgments:** The authors would like to thank Yan Ren, Julong He, Qiong Li, Yanzhang Weng, Wenwen Bai, Jiongwei Cao, Xiaomei Zhu, Beiming Kang, Weiwen Shen, Jie Zhao, Shoukai Cao, Peichong Pan, Jiahua Wan, Jia Wang and all other members of the Research Group from Tsinghua University and Qinghai University for contributions to the field experiment, without whom this work would not be possible.

**Conflicts of Interest:** The authors declare no conflict of interest.

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
