# Peer review of "In Situ Experimental Study of Cloud-Precipitation Interference by Low-Frequency Acoustic Waves"

_remotesensing, doi:10.3390/rs15040993_

Round 1
Reviewer 1 Report
This paper investigates the influences of the low-frequency acoustic waves on the cloud precipitation via in situ experiments. Comparing with the traditional methods, I think this an interesting method of the weather modification. The results and finding of this study can make contributions to the weather modification. However, the results of this method still have the challenges from the evaluation of the interference. Here are my concerns ans comments.
Lines 48-50, The authors address that the traditional methods can just cover limited area. However, the proposed method in this manuscript also can affect limited area too and highly depends on the operation site location,as the author said in the abstract (lines 27-29). So, I think the author can change a view to address the advantage of the new method.
Line 50-51. the author said “Meanwhile, the long-term use of 50 chemicals introduces ecological and environmental pollution risks. ” Is this statement supported by some references?
Please denote the relationship of the panel a, b, and c in the Fig.1.
Labels in Fig. 5 are too small. The captions of Fig. 6 is not clear enough, please refine it.
Some conclusions need more evidence, e.g, “the average Z value increases from 25.87 to 28.58 dBZ in the range of 100–500 m”, “ In general, the particle concentration increased significantly by 1–3 ln”. I agree that the low-frequency acoustic waves can impose effects on the cloud and precipitation. However, how can you quantify the effect of the operation from the natural change?
I suggest that the author can add some discussion between this method and the traditional method.
Author Response
The authors would like to thank the reviewer for their constructive comments and suggestions. The point-by-point responses to all comments are given below. The parts in italic are the reviewers’ comments, which are followed by our responses in blue.
Reviewer #1
This paper investigates the influences of the low-frequency acoustic waves on the cloud precipitation via in situ experiments. Comparing with the traditional methods, I think this an interesting method of the weather modification. The results and finding of this study can make contributions to the weather modification. However, the results of this method still have the challenges from the evaluation of the interference. Here are my concerns and comments.
[1] Lines 48-50, the authors address that the traditional methods can just cover limited area. However, the proposed method in this manuscript also can affect limited area too and highly depends on the operation site location, as the author said in the abstract (lines 27-29). So, I think the author can change a view to address the advantage of the new method.
Response: The authors would like to thank the reviewer for the helpful suggestion. Both conventional methods and acoustic operations have limited effects on clouds and precipitation and are highly dependent on the location of the operating site. The acoustic interference belongs to the ground operation, without the use of rockets and aircraft and other expensive carriers, overcoming the airspace restrictions of traditional operations.
[2] Line 50-51, The author said “Meanwhile, the long-term use of chemicals introduces ecological and environmental pollution risks.” Is this statement supported by some references?
Response: We agree with the reviewer’s comment. The references have all been checked and supplemented as appropriate. The silver ion (Ag+) in the catalyst AgI is a heavy metal that has some toxicity and may have a negative impact on the water environment. Since the 1960s [1], Ag+ concentrations in lakes and rivers [2,3], soils [4,5], and the atmosphere [6,7] after catalytic precipitation have been studied, but their toxic influences are still controversial [8,9]. This has been reflected in the revision on page 2.
[3] Please denote the relationship of the panel a, b, and c in the Fig. 1.
Response: Thanks for this suggestion. Panel (a) presents the location of the operation area in the source region of the Yellow River (SRYR), i.e., Darlag. Panel (c) shows the location of SRYR in the world. Panel (b) gives the distribution of rain gauges in the vicinity of the field site. This has been explained on page 4 in the revision.
[4] Labels in Fig. 5 are too small. The captions of Fig. 6 is not clear enough, please refine it.
Response: This has been done in the revised figures on pages 10-11 in the revised manuscript.
[5] Some conclusions need more evidence, e.g, “the average Z value increases from 25.87 to 28.58 dBZ in the range of 100–500 m”, “In general, the particle concentration increased significantly by 1–3 ln”. I agree that the low-frequency acoustic waves can impose effects on the cloud and precipitation. However, how can you quantify the effect of the operation from natural change?
Response: The authors very appreciate this insightful comment from the reviewer. The authors freely admitted that it is difficult to quantify the effects of acoustic manipulation from natural variability. This is because precipitation systems are constantly changing, developing and evolving, and there are no two identical clouds in nature. This indicates that the comparisons of rainfall measurements with and without acoustic treatment are not conducted under a completely fair background. The responses of macro- and microphysical properties of precipitation only represent acoustic interference effects coupled with natural variations in the Darlag region during the East Asian summer monsoon. In the future, additional field experiments with detailed monitoring of the cloud-precipitation process and numerical simulations with identical meteorological backgrounds should be performed to more comprehensively characterize acoustic intervention effects and obtain physical evidence of rain enhancement. This statement has been added to the conclusion of the revised manuscript to avoid misleading potential readers.
[6] I suggest that the author can add some discussion between this method and the traditional method.
Response: As suggested by the reviewer, the discussion between this method and the traditional method has been given in Supplementary Materials.
Operation method |
Features |
Keywords |
Manned aircraft |
Largest fixed investment, unclear operation area, inaccurate precipitation estimation, Fixed aircraft routes. Catalysts are required to be seeded in suitable clouds. Significantly affected by the ambient environment, droplet spectrum, and seeding height. |
Large area operation, large investment, airspace restrictions, chemical catalysis |
UAVs |
UAVs can operate in mountainous areas where manned aircraft have difficulty taking off and landing. The cost will be reduced when the amount of catalyst dispersed increases. |
Localized operation, precise and efficient, chemical catalysis |
Rockets, artillery |
The catalyst is driven into the clouds by vehicles such as rockets and artillery. The effect is influenced by the accuracy of the artillery shell striking the clouds, are highly influenced by wind speed and direction. |
Directional operation, mature operation process, uncertain effect, chemical catalysis |
Acoustic operation |
The acoustic waves with high energy are utilized to promote cloud droplet agglomeration. Relatively clear operating region with low cost. |
Directional operation, high mobility, physical catalysis |
Related references:
- Warburton, J.A.; Maher, C.T. The detection of silver in rainwater: Analysis of precipitation collected from cloud-seeding experiments. J. Appl. Meteorol. Climatol. 1965, 4, 560-564, doi: 10.1175/1520-0450(1965)004<0560:tdosir>2.0.co;2.
- Freeman, R. Ecological Kinetics of Silver in an Alpine Lake Ecosystem. In Proceedings of the Second Annual Symposium on Aquatic Toxicology, Morgan School of Biological Sciences, University of Kentucky, Lexington, KY, USA, 1979; pp. 342-358.
- Ćurić, M.; Janc, D. Wet deposition of the seeding agent after weather modification activities. Environ. Sci. Pollut. Res. 2013, 20, 6344-6350, doi:10.1007/s11356-013-1705-y.
- Williams, B.D.; Denhom, J.A. An assessment of the environmental toxicity of silver iodide-with reference to a cloud seeding trial in the snowy mountains of Australia. J. Weather Modificat. 2009, 41, 75-96, doi: 10.54782/jwm.v41i1.178.
- Fajardo, C.; Costa, G.; Ortiz, L.T.; Nande, M.; Rodriguez-Membibre, M.L.; Martin, M.; Sanchez-Fortun, S. Potential risk of acute toxicity induced by AgI cloud seeding on soil and freshwater biota. Ecotoxicol. Environ. Saf. 2016, 133, 433-441, doi: 10.1016/j.ecoenv.2016.06.028.
- Standler, R.B.; Vonnegut, B. Estimated possible effects of AgI cloud seeding on human health. J. Appl. Meteorol. Climatol. 1972, 11, 1388-1391, doi: 10.1175/1520-0450(1972)011<1388:epeoac>2.0.co;2.
- Causapé, J.; Pey, J.; Orellana-Macías, J.M.; Reyes, J. Influence of hail suppression systems over silver content in the environment in Aragón (Spain). I: Rainfall and soils. Sci. Total Environ. 2021, 784, 147220, doi: 10.1016/j.scitotenv.2021.147220.
- Jian, D.; Xiaofeng, L.; Hui, W.; Xueliang, G.; Jiming, L. Research progress on impact of AgI in weather modification operations on environment in related areas. Meteorol. Mon. 2020, 46, 257-268, doi: 1000-0526(2020)46:2<257:rgyxtq>2.0.tx;2-w.
- Korneev, V.P.; Potapov, E.I.; Shchukin, G.G. Environmental aspects of cloud seeding. Russ. Meteorol. Hydrol. 2017, 42, 477-483, doi: 10.3103/s106837391707007x.
- Shi, Y.; Wei, J.; Li, Q.; Yang, H.; Qiao, Z.; Ren, Y.; Ni, S.; He, J.; Shen, W.; Cao, S., et al. Investigation of vertical microphysical characteristics of precipitation under the action of low-frequency acoustic waves. Atmos. Res. 2021, 249, 105283, doi: 10.1016/j.atmosres.2020.105283.
- Shi, Y.; Wei, J.; Ren, Y.; Qiao, Z.; Li, Q.; Zhu, X.; Kang, B.; Pan, P.; Cao, J.; Wang, G. Investigation of precipitation characteristics under the action of acoustic waves in the source region of the Yellow River. J. Appl. Meteorol. Climatol. 2021, 60, 951–966, doi: 10.1175/jamc-d-20-0157.1.
- Wei, J.; Shi, Y.; Ren, Y.; Li, Q.; Qiao, Z.; Cao, J.; Ayantobo, O.O.; Yin, J.; Wang, G. Application of ground-based microwave radiometer in retrieving meteorological characteristics of Tibet Plateau. Remote Sens. 2021, 13, 2527, doi: 10.3390/rs13132527.

Author Response
The authors would like to thank the reviewer for their constructive comments and suggestions. The point-by-point responses to all comments are given below. The parts in italic are the reviewers’ comments, which are followed by our responses in blue.
Reviewer #2
This paper shows the cloud-precipitation interference technology based on acoustic agglomeration through field experiment. How to evaluate the effects or results of these cloud precipitation interference approaches, including the seed catalysts method, is still a problem since the atmospheric system is always changing. Also, it exists in this paper.
[1] There are some grammatical errors in the manuscript, for example line 59-60. Please recheck them.
Response: The authors should apologize for providing inadequate English expression. We carefully checked and corrected grammatical mistakes throughout the manuscript to comply with the technical writing specifications.
[2] Rename the section of 2, and 4. Especially for section 4, it is not discussion. And, add a new discussion section according to the journal's requirement.
Response: As suggested by the reviewer, the section names of 2 and 4 have been revised in the manuscript. Section 2 “Materials and methods” was renamed as “Experimental Setup”, and Section 4 “Discussion” was renamed as “Results and Discussion”.
[3] Analysis of experiment is superficial and it is needed for the expressions to be in focus. Please reorganize the conclusion section to highlight which aspects are affected by this interference method.
Response: As suggested by the reviewer, the conclusion section has been reorganized in the revision to emphasize the intervention influences of acoustic waves.
[4] To demonstrate the impacts of the interference method clearly, is it possible to set the experiment process as three step: no interference - interference - no interference?
Response: The authors agree with the reviewer. Based on a large number of pre-experiments to test the acoustic intervention effect on precipitation clouds, the duration of each acoustic operation was taken as 80 min, including one acoustic trial (40 min) and one control trial (40 min). That is, the acoustic device was turned on for 40 min and turned off during the next 40 min to represent the control group without acoustic application. A rainfall sample of at least 40 min before the acoustic operation was also obtained and used as a natural rainfall condition. That is, continuous 120-min monitoring was performed for overall rainfall observation. In Figures 10-12 in the manuscript, the three steps of no interference (nature condition) - interference - no interference (operation revocation) have been presented for better clarification. The authors should apologize that the duration of MRR- and MWR-derived samples was only 80 min due to deficient experimental design. The related statements have been supplemented in the experimental procedure on page 6.
The authors should also clarify that, although experimental procedure was divided into three stages in chronological order, a large number of randomized experiments were needed to validate the effects of the acoustic intervention. To date, it is indeed difficult to separate the acoustic effects from the natural evolution of precipitation by existing means. In the future, additional field experiments and more detailed monitoring of cloud-precipitation processes should be comprehensively performed to prove the effects of low-frequency acoustic waves on clouds and precipitation.
Related references:
- Warburton, J.A.; Maher, C.T. The detection of silver in rainwater: Analysis of precipitation collected from cloud-seeding experiments. J. Appl. Meteorol. Climatol. 1965, 4, 560-564, doi: 10.1175/1520-0450(1965)004<0560:tdosir>2.0.co;2.
- Freeman, R. Ecological Kinetics of Silver in an Alpine Lake Ecosystem. In Proceedings of the Second Annual Symposium on Aquatic Toxicology, Morgan School of Biological Sciences, University of Kentucky, Lexington, KY, USA, 1979; pp. 342-358.
- Ćurić, M.; Janc, D. Wet deposition of the seeding agent after weather modification activities. Environ. Sci. Pollut. Res. 2013, 20, 6344-6350, doi:10.1007/s11356-013-1705-y.
- Williams, B.D.; Denhom, J.A. An assessment of the environmental toxicity of silver iodide-with reference to a cloud seeding trial in the snowy mountains of Australia. J. Weather Modificat. 2009, 41, 75-96, doi: 10.54782/jwm.v41i1.178.
- Fajardo, C.; Costa, G.; Ortiz, L.T.; Nande, M.; Rodriguez-Membibre, M.L.; Martin, M.; Sanchez-Fortun, S. Potential risk of acute toxicity induced by AgI cloud seeding on soil and freshwater biota. Ecotoxicol. Environ. Saf. 2016, 133, 433-441, doi: 10.1016/j.ecoenv.2016.06.028.
- Standler, R.B.; Vonnegut, B. Estimated possible effects of AgI cloud seeding on human health. J. Appl. Meteorol. Climatol. 1972, 11, 1388-1391, doi: 10.1175/1520-0450(1972)011<1388:epeoac>2.0.co;2.
- Causapé, J.; Pey, J.; Orellana-Macías, J.M.; Reyes, J. Influence of hail suppression systems over silver content in the environment in Aragón (Spain). I: Rainfall and soils. Sci. Total Environ. 2021, 784, 147220, doi: 10.1016/j.scitotenv.2021.147220.
- Jian, D.; Xiaofeng, L.; Hui, W.; Xueliang, G.; Jiming, L. Research progress on impact of AgI in weather modification operations on environment in related areas. Meteorol. Mon. 2020, 46, 257-268, doi: 1000-0526(2020)46:2<257:rgyxtq>2.0.tx;2-w.
- Korneev, V.P.; Potapov, E.I.; Shchukin, G.G. Environmental aspects of cloud seeding. Russ. Meteorol. Hydrol. 2017, 42, 477-483, doi: 10.3103/s106837391707007x.
- Shi, Y.; Wei, J.; Li, Q.; Yang, H.; Qiao, Z.; Ren, Y.; Ni, S.; He, J.; Shen, W.; Cao, S., et al. Investigation of vertical microphysical characteristics of precipitation under the action of low-frequency acoustic waves. Atmos. Res. 2021, 249, 105283, doi: 10.1016/j.atmosres.2020.105283.
- Shi, Y.; Wei, J.; Ren, Y.; Qiao, Z.; Li, Q.; Zhu, X.; Kang, B.; Pan, P.; Cao, J.; Wang, G. Investigation of precipitation characteristics under the action of acoustic waves in the source region of the Yellow River. J. Appl. Meteorol. Climatol. 2021, 60, 951–966, doi: 10.1175/jamc-d-20-0157.1.
- Wei, J.; Shi, Y.; Ren, Y.; Li, Q.; Qiao, Z.; Cao, J.; Ayantobo, O.O.; Yin, J.; Wang, G. Application of ground-based microwave radiometer in retrieving meteorological characteristics of Tibet Plateau. Remote Sens. 2021, 13, 2527, doi: 10.3390/rs13132527.
Reviewer 3 Report
In current paper, the cloud-precipitation interference effect in situ based on a multi-dimensional multi-scale monitoring systems was evaluated. It is suggested to:
1.An outline of the paper at the end of the introduction section is recommended.
2. Did the authors compare the results of their study with the work of other authors?

3. The innovation of this paper is not highlighted. Explain clearly what the latest progress and previous work of this paper are based on?
4. All figure are well prepared and the resolution are great, except for figures 2 and 6. It is better to replace.
5. The application of the equations 1, 2 and 3 were not discussed. It is seriously suggested to discussed about the in the text,(How these equations were used in paper).
Author Response
The authors would like to thank the reviewer for their constructive comments and suggestions. The point-by-point responses to all comments are given below. The parts in italic are the reviewers’ comments, which are followed by our responses in blue.
Reviewer #3
In current paper, the cloud-precipitation interference effect in situ based on a multi-dimensional multi-scale monitoring systems was evaluated. It is suggested to:
[1] An outline of the paper at the end of the introduction section is recommended.
Response: This has been done in the revision.
[2] Did the authors compare the results of their study with the work of other authors?
Response: As suggested by the reviewer, this study was compared more with the findings of others.
[3] The innovation of this paper is not highlighted. Explain clearly what the latest progress and previous work of this paper are based on?
Response: The authors very appreciate this helpful comment from the reviewer. The innovation of this study has been highlighted on pages 2-3 in the revision. Research foundations and progress have been supplemented in the Introduction section of the manuscript.
[4] All figure are well prepared and the resolution are great, except for figures 2 and 6. It is better to replace.
Response: As suggested by the reviewer, Figures 2 and 6 have been provided at a sufficiently high resolution, at least 300 dpi.
[5] The application of the equations 1, 2 and 3 were not discussed. It is seriously suggested to discussed about the in the text. (How these equations were used in paper).
Response: Following this very good comment, the application of the mean difference (MD), normalized standard error (NSE) and Pearson correlation coefficient (CC) has been elaborated in the revision. The MD and NSE represent the degree of deviation of the same physical quantity measured by different devices. The CC value is used to quantify the correlation between the variables measured by the two devices. For multisource observations during the precipitation process, atmospheric and precipitation quantities such as the radar reflectivity factor (Z), rain rate (R), liquid water content (Il), droplet diameter (D), droplet number density (N), and cloud bottom height (Hc) are measured by different devices at the same time, and device consistency in different sampling spaces should be checked. Some of the devices have been examined in the authors' previous studies [10-12], and this study focuses on checking the consistency of R and Z. It is addressed to the referee only as personal communication but not reflected in the revised paper.
Table 1. Physical quantities for equipment consistency examination.
Parameters |
Sampling position |
Device - I |
Device - II |
Reflectivity factor, Z |
Near ground |
OTT |
MRR |
In the zenith direction |
MRR |
KaKu radar, in Ka band |
|
Rain rate, R |
Near ground |
OTT |
MRR |
Ground |
OTT |
Rain Gauges |
|
Liquid water content, Il |
In the zenith direction |
MRR |
MWR |
Droplet diameter, D |
Near ground |
OTT |
MRR |
Droplet number density, N |
Near ground |
OTT |
MRR |
cloud bottom height, Hc |
In the zenith direction |
MRR |
MWR |
Related references:
- Warburton, J.A.; Maher, C.T. The detection of silver in rainwater: Analysis of precipitation collected from cloud-seeding experiments. J. Appl. Meteorol. Climatol. 1965, 4, 560-564, doi: 10.1175/1520-0450(1965)004<0560:tdosir>2.0.co;2.
- Freeman, R. Ecological Kinetics of Silver in an Alpine Lake Ecosystem. In Proceedings of the Second Annual Symposium on Aquatic Toxicology, Morgan School of Biological Sciences, University of Kentucky, Lexington, KY, USA, 1979; pp. 342-358.
- Ćurić, M.; Janc, D. Wet deposition of the seeding agent after weather modification activities. Environ. Sci. Pollut. Res. 2013, 20, 6344-6350, doi:10.1007/s11356-013-1705-y.
- Williams, B.D.; Denhom, J.A. An assessment of the environmental toxicity of silver iodide-with reference to a cloud seeding trial in the snowy mountains of Australia. J. Weather Modificat. 2009, 41, 75-96, doi: 10.54782/jwm.v41i1.178.
- Fajardo, C.; Costa, G.; Ortiz, L.T.; Nande, M.; Rodriguez-Membibre, M.L.; Martin, M.; Sanchez-Fortun, S. Potential risk of acute toxicity induced by AgI cloud seeding on soil and freshwater biota. Ecotoxicol. Environ. Saf. 2016, 133, 433-441, doi: 10.1016/j.ecoenv.2016.06.028.
- Standler, R.B.; Vonnegut, B. Estimated possible effects of AgI cloud seeding on human health. J. Appl. Meteorol. Climatol. 1972, 11, 1388-1391, doi: 10.1175/1520-0450(1972)011<1388:epeoac>2.0.co;2.
- Causapé, J.; Pey, J.; Orellana-Macías, J.M.; Reyes, J. Influence of hail suppression systems over silver content in the environment in Aragón (Spain). I: Rainfall and soils. Sci. Total Environ. 2021, 784, 147220, doi: 10.1016/j.scitotenv.2021.147220.
- Jian, D.; Xiaofeng, L.; Hui, W.; Xueliang, G.; Jiming, L. Research progress on impact of AgI in weather modification operations on environment in related areas. Meteorol. Mon. 2020, 46, 257-268, doi: 1000-0526(2020)46:2<257:rgyxtq>2.0.tx;2-w.
- Korneev, V.P.; Potapov, E.I.; Shchukin, G.G. Environmental aspects of cloud seeding. Russ. Meteorol. Hydrol. 2017, 42, 477-483, doi: 10.3103/s106837391707007x.
- Shi, Y.; Wei, J.; Li, Q.; Yang, H.; Qiao, Z.; Ren, Y.; Ni, S.; He, J.; Shen, W.; Cao, S., et al. Investigation of vertical microphysical characteristics of precipitation under the action of low-frequency acoustic waves. Atmos. Res. 2021, 249, 105283, doi: 10.1016/j.atmosres.2020.105283.
- Shi, Y.; Wei, J.; Ren, Y.; Qiao, Z.; Li, Q.; Zhu, X.; Kang, B.; Pan, P.; Cao, J.; Wang, G. Investigation of precipitation characteristics under the action of acoustic waves in the source region of the Yellow River. J. Appl. Meteorol. Climatol. 2021, 60, 951–966, doi: 10.1175/jamc-d-20-0157.1.
- Wei, J.; Shi, Y.; Ren, Y.; Li, Q.; Qiao, Z.; Cao, J.; Ayantobo, O.O.; Yin, J.; Wang, G. Application of ground-based microwave radiometer in retrieving meteorological characteristics of Tibet Plateau. Remote Sens. 2021, 13, 2527, doi: 10.3390/rs13132527.

Round 2
Reviewer 1 Report
The author has responsed and refined all my concerns and comments. I have no additional comments.
Reviewer 2 Report
No more comments.